# Graphene Oxide–Platinum Nanoparticle Nanocomposites: A Suitable Biocompatible Therapeutic Agent for Prostate Cancer

**DOI:** 10.3390/polym11040733

**Published:** 2019-04-23

**Authors:** Sangiliyandi Gurunathan, Muniyandi Jeyaraj, Min-Hee Kang, Jin-Hoi Kim

**Affiliations:** Department of Stem Cell and Regenerative Biotechnology, Konkuk University, Seoul 05029, Korea; muniyandij@yahoo.com (M.J.); pocachippo@gmail.com (M.-H.K.)

**Keywords:** graphene oxide–platinum nanoparticles nanocomposites, prostate cancer, cytotoxicity, oxidative stress, mitochondrial membrane potential, DNA damage

## Abstract

Metal nanoparticles and the combination of metal nanoparticles with graphene oxide are widely used in environmental, agriculture, textile, and therapeutic applications. The effect of graphene oxide–green platinum nanoparticles (GO-PtNPs) on human prostate cancer cells (LNCaP) is unclear. Therefore, this study aimed to synthesize a nanocomposite of GO-PtNPs and evaluate their effect on prostate cancer cells. Herein, we synthesized GO-PtNPs using vanillin and characterized GO-PtNPs. GO-PtNP cytotoxicity in LNCaP cells was demonstrated by measuring cell viability and proliferation. Both decreased in a dose-dependent manner compared to that by GO or PtNPs alone. GO-PtNP cytotoxicity was confirmed by increased lactate dehydrogenase release and membrane integrity loss. Oxidative stress induced by GO-PtNPs increased malondialdehyde, nitric oxide, and protein carbonyl contents. The effective reactive oxygen species generation impaired the cellular redox balance and eventually impaired mitochondria by decreasing the membrane potential and ATP level. The cytotoxicity to LNCaP cells was correlated with increased expression of proapoptotic genes (p53, p21, Bax, Bak, caspase 9, and caspase 3) and decreased levels of antiapoptotic genes (Bcl2 and Bcl-xl). Activation of the key regulators p53 and p21 inhibited the cyclin-dependent kinases Cdk2 and Cdk4, suggesting that p53 and p21 activation in GO-PtNP-treated cells caused genotoxic stress and apoptosis. The increased expression of genes involved in cell cycle arrest and DNA damage and repair, and increased levels of 8-oxo-deoxyguanosine and 8-oxoguanine suggested that GO-PtNPs potentially induce oxidative damage to DNA. Thus, GO-PtNPs are both cytotoxic and genotoxic. LNCaP cells appear to be more susceptible to GO-PtNPs than to GO or PtNPs. Therefore, GO-PtNPs have potential as an alternate and effective cancer therapeutic agent. Finally, this work shows that the combination of graphene oxide with platinum nanoparticles opens new perspectives in cancer therapy. However further detailed mechanistic studies are required to elucidate the molecular mechanism of GO-PtNPs induced cytotoxicity in prostate cancer.

## 1. Introduction

The mortality rate of cancer is increased rapidly by both aging and growth of the population and is also associated with socioeconomic development [1]. According to the International Agency for Research on Cancer, it is estimated 18.1 million new cancer cases and 9.6 million cancer deaths in 2018. Prostate cancer is the second most common cancer in men and fourth most commonly occurring cancer overall and most frequently diagnosed cancer, which is occurs one in nine in older man [1,2]. Prostate cancer can be treated by conventional therapies such as radiation, chemo, hormone, cryo and surgery; however, the treatment is depends on individual cases. Chemo-drugs, such as docetaxel, cabazitaxel, mitoxantrone and estramustine, are used to treat prostate cancer, however they cause undesired side effects. Another major cause of radiotherapy failure is the radioresistance of certain cancers. Therefore, finding alternative, biocompatible treatments is necessary.

Nanomedicine has been proposed as a new tool and alternative for cancer therapy and diagnosis. Recently, biomolecule mediated synthesis of metal nanoparticles shows great interest and rapidly used both academic and medical industry aspects due to their unique properties and promising applications as catalysts, ferrofluids, and semiconductors [3,4]. Several metal nanoparticles, such as silver, gold, and palladium, were synthesized using biomolecules, such as bacterial extracts, fungi extracts, plant extracts, and purified phenolic compounds, and tested for cytotoxicity against various types of cancer cells including human breast cancer cells, lung cancer cells, human ovarian cancer cells, and neuroblastoma cancer cells [5,6,7,8]. Small metallic nanoparticles seem to be potential nanodrugs to optimize the performances of radiotherapy. Among other nanomaterial systems, platinum nanoparticles (PtNPs) with radiation are used as radiation dose enhancers and anticancer drug carriers in cancer therapy. For example, Porcel et al. [9] developed a new strategy based on the combination of platinum nanoparticles with irradiation by fast ions effectively used in hadron therapy. The results demonstrated that PtNPs strongly enhance lethal damage in DNA, with an efficiency factor close to 2 for double-strand breaks. Platinum complexes and platinum NPs (PtNPs) have shown excellent properties to amplify radiation effects [9,10,11].

Since several years, platinum-based drug molecules have received much attention due to their electro-catalytic properties. For instance, platinum-based therapeutic drugs, notably cisplatin and carboplatin, have been exploited in chemotherapy to kill cancer cells [12]. However, these drugs do not have specificity towards cancer cells and have effects on normal cells leading to substantial dose-limiting acute and chronic toxicities. Since undesired toxic side effects and frequent development of drug resistance represent the major challenges in cancer therapy, it is therefore necessary to develop cisplatin analogs or other metal complexes that are able to offer a more acceptable level of toxicity and improved antitumor activity [13]. 

Graphene-based nanocomposites with metal nanoparticles show immense interest due to their extraordinary physical, chemical, and biological properties [14,15]. The excellent properties of combination of graphene based materials and metal nanoparticles shows promising nanomaterial in many fields of application such as electronic-devices, sensors, nanocomposites, energy storage, and supercapacitors [16,17,18,19]. Metal nanoparticles, such as silver and graphene oxide, exhibit significant antibacterial activity against Gram-negative and -positive bacteria and anticancer activity against human ovarian cancer cells and human neuroblastoma cancer cells [8,20,21]. In particular, PtNPs are highly important metallic catalysts for many electrochemical reactions and used as sensors towards biological and drug molecules [22,23]. The combination of rGO and PtNPs exhibits promising electrocatalytic activity and selectivity in the detection of target molecules [24]. The major advantage of using graphene oxide for the preparation of nanocomposites, such as graphene oxide–platinum nanoparticles, is a cheap and accessible nanomaterial with abundant oxygen-containing functional groups, which is indispensable to anchor novel metal ions [25,26]. Furthermore, Pt ions are easily absorbed uniformly by GO due to the presence of abundant hydroxyl and carboxyl functional groups. Wu et al. [27] fabricated reduced graphene oxide (RGO)/metal (oxide) composites using glucose as the reducing agent and the stabilizer. The developed composite nanomaterials show excellent electrode catalyst to simultaneous electrochemical analysis of l-ascorbic acid, dopamine, and uric acid. Ali et al. [28] synthesized variety of composites by simultaneous reduction of variety of nanoparticles, such as palladium, platinum, silver, and gold with graphene oxide using a black pepper extract (BPE) for quantification and kinetic analysis of epidermal growth factor receptor (ErbB2), for application to breast cancer diagnostics. The developed composites exhibited less toxic, biocompatible, and antioxidants properties, and can detect low concentrations of ErbB2. A nanocomposite consisting of combination of various nanomaterials such as reduced graphene oxide combined with manganese-doped zinc sulfide quantum dots and functionalized with folic acid (FA-rGO/ZnS:Mn) and loaded with doxorubicin (DOX). DOX is adsorbed on the surface of graphene sheets and releases efficiently with specificity, against folate-rich breast cancer cells, which is the best platform for targeted cancer treatment [29]. Single-crystal metastable manganese sulfide nanowires (γ-MnS NWs) coated with graphitic carbon exhibited brittle with a Young’s modulus of 65 Gpa show the cycling stability of stable microsized α-MnS, with an initial capacity of 1036 mAh g^−1^ and a reversible capacity exceeding 503 mAh g^−1^ after 25 cycles [30]. A novel regorafenib (REG) electrochemical sensor, developed by reduction of a zirconia-nanoparticle and reduced graphene oxide composite (ZrO_2_/rGO/GCE) using hydrazine hydrate as reducing and stabilizing agent, shows an excellent electrocatalytic response and determination of REG in the presence of ascorbic acid and uric acid at lower concentration in serum samples [31]. The PtNPs were found to cause DNA strand breaks in a concentration-, time-, and size-dependent manner [32], and inhibit DNA replication, whereas the rich oxygen-containing functional groups of graphene oxide on the surface provide it with the opportunity to be modified by many functional molecules to expand biological applications and reduce toxicity. Combination of graphene oxide with nanoparticles in a single platform could provide the simultaneous administration of two or more active agents that are known to disrupt multiple targets, resulting in a more efficient solution to cancer or any other treatments with two different physical and chemical properties. The combination of graphene oxide and platinum nanoparticles could provide efficient synergistic effect on cytotoxicity compared to its counterpart.

Although several studies have reported that the effect of combination of graphene oxide with various metal nanoparticles, such as silver, gold, palladium, etc., against various type of cancer cells, so far there is no report about the anticancer effect of graphene oxide–platinum nanoparticles nanocomposites. This is the first study aimed to address the following objectives including synthesis of graphene oxide–platinum nanoparticles nanocomposite using vanillin as reducing agent. The second objective is to address the cytotoxic effect of graphene oxide–platinum nanoparticles nanocomposite in prostate cancer cells. The final objective is to address the mechanism of anticancer effect of graphene oxide–platinum nanoparticles nanocomposite in prostate cancer cells using various cellular assays.

## 2. Materials and Methods 

### 2.1. Synthesis and Characterization of GO, PtNPs, and GO-PtNPs

Graphene sheets were synthesized by Hummers’ method with slight modification as required [8,33]. Synthesis and characterization of the GO-PtNP nanocomposite was done as described previously [20]. Detailed materials and methods are given in the Appendix A. 

### 2.2. Cell Viability and Cell Proliferation Assay

The cells were grown to logarithmic growth phase and mixed with various concentrations of GO, GO-PtNPs, and PtNPs for 24 h, followed by determinations of cytotoxicity. The inhibitory concentration (IC50) is defined as the concentration of PtNPs causing 50% inhibition of growth of the cells. Cell proliferation was determined using bromodeoxyuridine/5-bromo-2′-deoxyuridine (BrdU) (Roche, Basel, Switzerland). 

### 2.3. Measurement of Cytotoxicity and Cell Mortality

The membrane integrity of LNCaP cells was evaluated using an LDH Cytotoxicity Detection Kit (Sigma-Aldrich, St. Louis, MS, USA) according to the manufacturer’s instructions. Cell mortality was evaluated using the trypan blue assay as described previously [7]. 

### 2.4. Determination of Reactive Oxygen Species (ROS), Malondialdehyde (MDA), Nitric Oxide (NO), and Carbonylated Protein Levels 

Reactive oxygen species (ROS) were estimated as described previously [34]. Briefly, LNCaP cells were seeded into wells of 24-well-plates at a density of 5 × 10^4^ cells per well and cultured for 24 h. MDA levels were determined using a thiobarbituric acid reactive substances assay as previously described with suitable modifications [34]. NO production was quantified spectrophotometrically using Griess reagent (Sigma-Aldrich, St. Louis, MO, USA). Absorbance was measured at 540 nm and nitrite concentration was determined using a calibration curve prepared with sodium nitrite as the standard [35]. Carbonylated protein content was measured according to Uehara and Rao [36].

### 2.5. Measurement of Mitochondrial Membrane Potential (MMP) and ATP Level

MMP was measured according to the manufacturer’s instructions (Molecular Probes, Eugene, OR, USA) using the cationic fluorescent indicator, JC-1 (Molecular Probes). The ATP level was measured according to the manufacturer’s instructions (Catalog Number MAK135; Sigma-Aldrich, St. Louis, MO, USA). 

### 2.6. Measurement of Antioxidative Marker Levels

The expression levels of oxidative and antioxidative stress markers were measured as described previously [37]. 

### 2.7. Measurement of 8-oxo-dG and 8-oxo-G

8-oxo-dG was determined as described previously [38] and using the manufacturer’s instructions (Trevigen, Gaithersburg, MD, USA).

### 2.8. Statistical Analysis

All assays were conducted in triplicate, and each experiment was repeated at least three times. The results are presented as the means ± standard deviation. All experimental data were compared using Student’s *t*-test. A *p*-value < 0.05 was considered statistically significant. Results are expressed as mean ± standard deviation of three independent experiments. There was a significant difference in treated cells compared to untreated cells with Student’s *t*-test (* *p* < 0.05).

## 3. Results and Discussion

### 3.1. Synthesis and Characterization of GO and GO-PtNP by UV-visible Spectroscopy 

The ultraviolet–visible spectrum of synthesized GO particles exhibited two characteristic absorption peaks at 230 nm, which can be attributed to the π–π* transition of aromatic C=C bonds, and a shoulder at 300 nm, corresponding to the n–π * transition of C=O bonds [39]. The hydrophilic property of the oxygenated graphene layers imparts significant solubility and stability in water. The absorption peak for GO-PtNPs was red-shifted to 267 nm (Figure 1A,B), owing to the restoration of sp^2^ carbon atoms. This characteristic red-shift is considered a monitoring tool for the graphene–platinum nanoparticle nanocomposite [8,20].

### 3.2. FTIR Analysis of GO and GO-PtNPs

The synthesis of GO from native graphite and its decoration with PtNPs were analyzed by Fourier-transform infrared (FTIR) spectroscopy. The FTIR spectra of GO and the GO-PtNP composite are shown in Figure 2A,B. The spectrum of GO (Figure 2A) showed a strong and broad band at 3300 cm^−1^ due to the –OH stretching vibration. The carbonyl (–C=O) stretching of carboxylic groups present at the edge planes of the GO sheets was observed at 1730 cm^−1^. The absorption due to –OH bending, epoxide groups, and skeletal ring vibrations were observed at 1600 cm^−1^. After decoration of PtNPs on the surface of GO, the –OH stretching vibration and carbonyl (–C=O) stretching of carboxylic groups were shifted to 3320 and 1725 cm^−1^, respectively. Interestingly, the deformation stretching frequency of –OH groups attached to the aromatic ring was 1380 cm^−1^ [40]. The peaks were observed in the spectrum of GO-PtNPs at 1725 and 1650 cm^−1^ corresponding to C=O stretching vibrations of COOH groups, which were attributed to C=O bonds in the carboxylic acid and carbonyl moieties, respectively (Figure 2B), and another strong peak appears at 1150 indicating C–OH stretching. All these data confirmed the formation of GO from native graphite, generation of oxygen-containing functionalities during oxidation process, and decoration of PtNPs on the surface of GO. These observations agreed with those reported in the literature [41,42]. The collect data suggested that the vanillin, aphenolic compound is responsible for synthesis of PtNPs and decoration of PtNPs on the surface of GO.

### 3.3. X-Ray Diffraction Analysis of GO and GO-PtNPs

X-ray diffraction (XRD) was performed to confirm the formation structures of GO and GO-PtNPs. Figure 3A,B display the XRD patterns of GO and GO-PtNPs. The diffraction peak of GO was observed at 11.5, corresponding to the (200) plane and an interlayer distance of 0.76 nm [19,38]. The newly appeared diffraction peaks located at 39.8, 46.5, 55.0, and 70.6 corresponded to the (111), (200), (220), and (311) crystal planes of Pt, respectively (JCPDS No. 01-087-0646), demonstrating that PtNPs were decorated uniformly on the GO surface [19,43], and confirming the presence of platinum particles on the graphene substrate. The average size of the PtNPs was calculated to be 2 nm using the Scherrer equation based on the full width at half maximum of the Pt (111) diffraction peak. The findings strongly agreed with those of previously published reports [19,42,43]. After chemical oxidation, the (002) peak of graphite was shifted to 11.5° with a d spacing of 0.94 nm. This shift might be attributed to the introduction of oxygen-containing functional groups like epoxy, hydroxyl, carbonyl, and carboxylic groups at both the sides and edges of the graphene sheets. This confirmed the formation of GO from natural graphite during the oxidation process and the formation of platinum particles [42]. 

### 3.4. Raman Spectroscopy Analysis of GO and GO-PtNPs

Raman spectroscopy was used to investigate the structure of GO and GO-PtNPs. The main features in the Raman spectra of graphitic carbon-based materials are the G and D peaks and their overtones [44]. Figure 4A,B shows the Raman spectra of the prepared GO and GO-PtNPs. The two most intense peaks were the D and G band at 1360 and 1580 cm^−1^, respectively. A prominent 2D band at 2690 cm^−1^ was evident, and a defect-activated peak (D + G) was also visible at approximately 2950 cm^−1^. The D peak represents first order resonance and the breathing mode of aromatic rings arising due to the defects in the sample. The D peak intensity is used to measure of the degree of disorder [45]. The G peak is due to the bond stretching of all pairs of sp^2^ atoms in both the rings and chains and corresponds to the optical E2g phonons. The intense D peak along with a large bandwidth suggested the significant structural disorder in GO. The 2D peak at approximately 2690 cm^−1^ is attributed to double resonance transitions resulting in the production of two phonons with opposite momentum. Simultaneously, the ratio of the intensity of the D band to the G band (ID/IG) increased from 1.49 to 1.89 between GO and PtNPs decorated GO.

### 3.5. Morphology and Size Analysis of GO and GO-PtNPs using SEM and TEM 

The morphology of GO and GO-PtNPs was analyzed by scanning electron microscopy. Micrographs of GO (Figure 5A) revealed a two-dimensional sheet-like structure consisting of multiple lamellar layers. The edges of individual sheets were visible, as has been described [8,46,47]. GO-PtNPs images (Figure 5B) revealed that the PtNPs particles were uniformly distributed throughout the graphene layers without any agglomeration in the substrate on the surface of GO, which agreed with previously published reports of other nanoparticles, such as silver [8]. Field emission scanning electron microscopy images revealed well-decorated PtNPs on the surface of GO nanosheets. Smaller PtNPs were spherical in shape, whereas the bigger particles had an elongated form. This elongated shape was attributed to an agglomeration of the highly concentrated PtNPs. The size of GO and PtNPs decorated GO were characterized by transmission electron microscopy. As shown in Figure 5C,D, transparent and wrinkled layers of GO nanosheets were observed. After successful decoration, PtNPs decorated on the graphene oxide layers were spherical with a diameter < 2 nm, which is consistent with the XRD results, confirming the formation of GO-PtNP nanocomposites. The results indicated that the combination of graphene oxide and PtNPs suggest that graphene oxide facilitates the reshaping and coarsening of PtNPs during simultaneous reduction of graphene oxide and PtNPs. The reducing and stabilizing agents promoted biomolecule induced transformations of nanohybrids (Appendix A).

### 3.6. Effect of GO-PtNPs on Viability of LNCaP Cells 

The effect of GO-PtNPs on LNCaP androgen-sensitive human prostate adenocarcinoma cells was explored to evaluate the approach in the treatment of prostate cancer. To evaluate and optimize the dose, and explore a dose-dependent effect, LNCaP cells were treated with various concentrations of GO (20–100 µg/mL), GO-PtNPs (5–25 µg/mL), and PtNPs (10–50 µg/mL) for 24 h. The three compounds displayed dose-dependent cytotoxicity on LNCaP cells. Interestingly, GO-PtNPs showed effective responses on cell viability compared to control, parental GO, and PtNPs. GO-PtNPs nanocomposites exhibited significant cytotoxicity against the human prostate cancer cells and the cytotoxicity was greater as the concentration increased. GO did not affect cell viability as drastically. Even at very high concentration (100 µg/mL), the loss of viability was only 60%. This low cytotoxicity of GO was probably attributed to the enrichment of oxygen atoms on the surface of GO in the form of carboxyl, epoxy, and hydroxyl groups, which reduces cell toxicity [48]. The loss of viability of PtNPs was dose-dependent and slightly better than GO, and less than GO-PtNPs. Even at a very high concentration (50 µg/mL), the loss of viability was only 80%, whereas at a high concentration of GO-PtNPs (25 µg/mL), the loss of viability was 99%. The findings indicated that the GO-PtNPs nanocomposite had enhanced anticancer capability, in contrast to the relatively low toxicity of GO and PtNPs on LNCaP cells. Collectively, the data indicated a dose-dependent inhibition of the cell viability with GO in the range of 20 to 100 μg/mL, with an IC50 of approximately 80 μg/mL (Figure 6A). Using GO-PtNPs, a dose-dependent inhibition of the cell viability was observed in the range of 5 to 25 μg/mL with an IC50 of approximately 10 μg/mL (Figure 6B). With PtNPs, a dose-dependent inhibition of the cell viability was observed in the range of 10–50 μg/mL with an IC50 of approximately 30 μg/mL (Figure 6C). Thus, GO-PtNPs displayed a more pronounced inhibitory effect on cell viability than the other tested nanomaterials and represent a promising candidate for treatment of prostate cancer cells. Similarly, GO-AgNPs nanocomposites effectively inhibit cell viability of a variety of bacteria, human ovarian cancer cells, and cervical cancer cells [8,20,49,50,51]. Further, the cell morphology of GO, GO-PtNPs, and PtNPs was examined phase-contrast microscope, all the treated cells compromised cell structure, and cells were round in shape and all the dead cells were detached from surface. The effect on cell viability was clearly dose-dependent (Appendix A). 

### 3.7. GO-PtNPs Inhibit Proliferation of LNCaP Cells

To determine the antiproliferative action of GO-PtNPs on LNCaP cells, the cells were treated with various concentrations of GO, GO-PtNPs, and PtNPs for 24 h, and proliferation was determined using BrdU. The growth rates of prostate cancer cells treated with GO-PtNPs were significantly decreased compared to that of the control, and similar results were observed in LNCaP cells with GO and PtNPs treatment. However, the effective inhibition of proliferation was observed with GO-PtNPs compared to control, GO, and PtNPs (Figure 7A–C). Of note, GO-PtNPs exhibited higher antiproliferative action, which was comparable to the parental GO and PtNPs. The data suggested that GO-PtNPs suppress proliferation. Similarly, goserelin-loaded nanoparticles influence the growth of LNCaP prostate cancer cells by the direct induction of necrosis and apoptosis [52].

### 3.8. GO-PtNPs Induce Cytotoxicity in LNCaP Cells

Membrane integrity determines the fate of cells and is an important factor for cell survival. Membrane integrity can be estimated by measuring leakage of LDH, which is a cytosolic enzyme that aids in the conversion of lactate to pyruvate. When membrane integrity is compromised, the enzyme is secreted. To determine the leakage of LDH, LNCaP cells were treated with IC50 concentrations of GO, GO-PtNPs, and PtNPs for 24 h. GO-PtNPs comparatively significantly increased the leakage of LDH compared to that in the control cells (Figure 8A). GO and PtNPs also increased the leakage of LDH to a greater extent than that in the control group. However, leakage was lower than that resulting from treatment with GO-PtNPs, indicating that the combination of GO and PtNPs acted synergistically to induce cytotoxicity. GO produces moderate leakage of LDH in a variety of cancer cells including human breast cancer cells [47], human ovarian cancer cells [8], and human cervical cancer cells [51]. One study [53] reported that human lung cancer cells treated with silver and platinum nanoparticles released significantly greater amounts of LDH compared to the control. The present and previous data suggest a correlation between cell viability and damage to the membrane caused by the nanoparticles, which ultimately proves lethal [54]. Membrane integrity and cell survival was examined by the trypan blue exclusion assay. Normal healthy cells are able to exclude the dye, but trypan blue will diffuse into cells in which membrane integrity has been lost. LNCaP cells were treated with IC50 concentrations of GO, GO-PtNPs, and PtNPs for 24 h. Significant cytotoxic effect was consistently observed in LNCaP cells. Cytotoxicity was most pronounced for cells treated with GO-PtNPs compared to that for treatments with GO and PtNPs (Figure 8B).

### 3.9. GO-PtNPs Increase the Level of Oxidative Stress Markers

The effect of GO-PtNPs on ROS generation was evaluated using DCFH2-DA. LNCaP cells were treated with IC50 concentration of GO, GO-PtNPs, and PtNPs for 24 h and then cells were exposed to 40 μM DCFH2-DA for 30 min. First, we measured the distribution of the fluorescence intensity in the presence or absence of GO, GO-PtNPs and PtNPs. The treatment of LNCaP cells with GO-PtNPs led to a marked shift to greater fluorescence peak intensities compared to the untreated control (Figure 9). Moreover, ROS generation induced by GO-PtNPs resulted in high FITC fluorescence intensity, indicating an increased susceptibility to oxidative stress. GO-PtNP-treated cells were more than four times more susceptible than the control samples, while cells treated with GO and PtNPs exhibited susceptible than the control sample (Appendix A); these results are consistent with the inhibitions of growth and cell proliferation. Subsequently, we evaluated the level of MDA in GO-, GO-PtNP-, and PtNP-treated cells. The MDA level was significantly higher in GO-PtNP-treated cells than either GO or PtNPs treatments (Figure 9B). Lipid peroxidation is a process of oxidation of polyunsaturated fatty acids due to the presence of several double bonds in their structure and it involves production of peroxides, ROS, and other reactive species, such as MDA. MDA is a reactive byproduct of lipid peroxidation and an end product that interacts with DNA to forming 3-(2-deoxy-β-d-erythro-pentofuranosyl)pyrimido[1,2-α]purin-10(3H)-one adducts [55]. Increased lipid peroxidation and decreased GSH were reported in human embryonic kidney (HEK)293 cells exposed to PtNPs [56].

In general, nanoparticles, such as silver, platinum, and palladium ions, bind to protein disulfide bonds in the cytoplasm, causing deformities in the protein structure. These malformed proteins are then incorporated into the plasma membrane, leading to alterations in cell permeability and cellular death [7,47]. In addition, graphene induces toxicity owing to its distinct physicochemical characteristics such as purity, lateral dimension, size of the sheets, and oxidation state, which may influence its cellular uptake, biodegradation, and toxicity. Once in contact with the cell membrane, graphene sheets can create an impermeable encasement affecting the normal exchange between the cell and the extracellular environment, graphene oxide can also damage the cell membrane through strong electrostatic interactions between the negatively charged oxygen groups on its surface and the positively charged lipids present on cell membranes [57,58,59,60,61,62].

Nanocytotoxicity may be caused by the induction of oxidative and/or nitro-oxidative stress [63,64]. Therefore, we were interested to determine the fate of GO-PtNPs on the generation of reactive nitrogen species (RNS). LNCaP cells were treated with IC50 concentrations of GO, GO-PtNPs, and PtNPs for 24 h and then the level of nitric oxide (NO) was determined. Cells treated with either GO or PtNPs displayed an 8-fold higher level of NO than control. Interestingly, the level of NO was significantly higher in all tested groups relative to control. The increase in the NO level after 24-h treatment with GO-PtNPs was 18-fold higher than after incubation with the same concentration of GO or PtNPs (Figure 9C). Overproduction of ROS and RNS in cells influences various cellular processes and components, promotes DNA breakage, and impairs the antioxidant potential; it has also been associated with carcinogenesis [65]. An increased level of NO in adenocarcinoma cells treated with silver nanoparticles (AgNPs) alone and in combination with an inhibitor of histone deacetylase (HDAC) was reported [66].

Carbonyl groups appear to be a significant and stable marker of the oxidative stress that results from the oxidation of proteins. Increased oxidative stress during altered homeostasis of prooxidants and antioxidants leads to deleterious effects on cellular components through oxidative damage to proteins, lipids, and nucleic acids. LNCaP cells were treated with IC50 concentrations of GO, GO-PtNPs, and PtNPs for 24 h and then the level of carbonylated protein was determined. Cells treated with GO, PtNPs, and GO-PtNPs showed 5-fold, 8-fold, and 12-fold higher level, respectively, of carbonylated protein than control (Figure 9D). 

### 3.10. GO-PtNPs Decrease MMP

The integrity of the mitochondrial membrane is regulated by its membrane potential, which influences electron transport and oxidative phosphorylation. Alteration of MMP causes cellular apoptosis. Spectrophotometric analysis showed that MMP was significantly compromised in GO-PtNP-treated cells compared with the control, and treatment with GO or PtNPs decreased the MMP, and the decrease was markedly more pronounced using GO-PtNPs. Twenty-four hours of treatment with GO-PtNPs increased the percentage of cells with low MMP (Δ*ψ*m) compared to that of the untreated cells (Figure 10A). The effect of GO-PtNPs on the MMP of LNCaP cells was evaluated using fluorescence microscopy. LNCaP cells incubated with GO, GO-PtNPs, and PtNPs underwent mitochondrial damage, resulting in changes in the ΔΨm. Consequently, the JC-1 aggregate level was significantly decreased by GO-PtNPs compared to GO or PtNPs, which resulted in low FITC fluorescence intensity, indicating an increased susceptibility to oxidative stress (Appendix A). Similarly, others [67] reported that a buffalo rat liver cell line treated with silicon oxide nanoparticles displayed increased cytotoxicity and mitochondrial damage accompanied by decreases in mitochondrial dehydrogenase activity, MMP, enzymatic expression in the Krebs cycle, and activity of the mitochondrial respiratory chain complexes I, III, and IV. Similarly, another study [34] reported the decreased level of MMP and ATP in the presence of AgNPs alone and in combination with an HDAC inhibitor.

Next, we examined the impact of GO-PtNPs on ATP synthesis. Synthesis of ATP is highly dependent on the integrity of the mitochondrial membrane, which regulates the pumping of hydrogen ions across the inner membrane during electron transport and oxidative phosphorylation [67]. Therefore, the effect of GO-PtNPs on the ATP synthesis in LNCaP cells was evaluated. As shown in Figure 10B, compared with that in the control, treatment of LNCaP cells with GO, PtNPs, and GO-PtNPs resulted in a decreased level of ATP, but these decreases were highly significant in the latter treatment. These results indicated that ATP synthesis was suppressed by GO-PtNPs. A 24-h treatment with GO-PtNPs resulted in decreased ATP synthesis with low MMP (Δ*ψ*m), compared to that in the untreated cells. This result suggested that MMP and ATP synthesis are associated. 

### 3.11. GO-PtNPs Impair Antioxidant Systems

The generation of ROS in cells occurs in equilibrium with a wide variety of antioxidant molecules, including SOD, catalase, GPx, and peroxiredoxins, as well as nonenzymatic scavengers, such as vitamin C, vitamin E, GSH, lipoic acid, carotenoids, and iron chelators [68]. Therefore, we evaluated the impact of oxidative stress in GO-PtNP-induced prostate cancer cell death. LNCaP cells were treated with IC50 concentrations of GO, GO-PtNPs, and PtNPs for 24 h and the protein level of selected antioxidant enzymes was determined (GSH, GSH, GSSG, SOD, CAT, GPx, and TRX). A statistically significant reduction in the levels of all enzymes was evident after treatment with GO, PtNPs, and GO-PtNPs. The latter treatment produced the greatest reductions (Figure 11). A previous study reported that AgNPs also reduced the activities of GSH, GSH: GSSG, SOD, CAT, GPx, and TRX [38]. Similarly, a reduced level of SOD was observed in human skin carcinoma and human fibrosarcoma after exposure to 7–20 nm AgNPs [69,70]. Decreased GPX activity in rat pheochromocytoma and mouse neuroblastoma cells by ZrO_2_NPs < 100 nm was reportedly related to genotoxic and cytotoxic effects [71]. Others [72] observed that PANC-1 cells treated with 2.6 nm and 18 nm AgNPs displayed decreased levels of SOD1 protein and mRNA, respectively. The lower level of SOD activity in MCF-7 breast cancer cells leads to a drastic alteration in the morphology of the mitochondria associated with increased fragmentation and swelling of the matrix [73]. Consistent with previous results from GO-PtNP-treated LNCaP cells [73], the cell death that occurred following the reduction of SOD level by SOD inhibitor likely occurred through a combination of the regulated mechanism (apoptosis) and unregulated mechanism (oxidative damage to the organelles). The collective data indicate that GO-PtNPs effectively influence the level of antioxidant molecules and eventually compromise the redox balance in LNCaP cells. 

### 3.12. Effect of GO-PtNPs on Expression of Proapoptotic and Antiapoptotic Genes

Reduced ROS is considered an essential regulator of the normal physiological functions of cells. Increased levels of ROS could damage proteins, nucleic acids, lipids, membranes, and organelles, which can lead to activation of cell death processes like apoptosis [74]. ROS play significant roles in the activation of various cellular signaling pathways and transcription factors. One of the most targeted genes with respect to DNA damage are tumor suppressor p53, a gene involved in cell cycle arrest, DNA repair, senescence, and apoptosis [75,76], and p21, a gene involved in cell cycle regulation. Furthermore, ROS induces the activity of genes, such as Bax and Bak, which are involved in mitochondria mediated apoptosis and downregulates Bcl2 and Bcl-xl. To determine the effect of GO-PtNPs on expression of proapoptotic and antiapoptotic genes, cells were treated with IC50 concentrations of GO, GO-PtNPs, and PtNPs for 24 h, and the m-RNA expressions of the target genes were determined. As expected, the levels of p53, p21, Bax, Bak, caspase-9, and caspase-3 were significantly upregulated, whereas the levels of Bcl2 and Bcl-xl were significantly downregulated by up to three-fold (Figure 12). Bcl-2 expression decreased 2.5 times compared to the control. A previous study [77] found that cells treated with PtNPs experienced genotoxic stress due to the activation of p53 and p21, which eventually led to proliferating cell nuclear antigen-mediated growth arrest and apoptosis. Our results were consistent with the recent finding [78] that PtNP-treated DEN animals showed significant increase in liver p53 gene expression level than normal rats. Other [56] reported that exposure of HEK293 cells to PtNPs induced the upregulation of Bax and the downregulation of Bcl2. 

The findings confirm that PtNPs induce mitochondria-mediated apoptosis, which is primarily responsible for cisplatin production and ROS generation in cell organelles [79]. Caspases play a significant role in apoptosis. We found that the LNCaP cells treated with IC50 concentrations of GO, GO-PtNPs, and PtNPs for 24 h displayed significantly increased caspase-9 and caspase-3 activities. Caspase-9 plays a significant role in signal transduction by induction of the executioner caspases-3 and -7 [38,80]. Similarly, PtNPs induced caspase-3 activity in a dose-dependent manner in HEK293 cells [56]. Cisplatin-induced DNA damage is related to the ratio of proapoptotic and antiapoptotic proteins, and release of cytochrome c from mitochondria followed by the activation of cysteine caspases selectively degrades the target proteins [81]. The collective data indicate that GO-PtNPs induce the intrinsic pathway of apoptosis, which is mediated by the upregulation of proapoptotic genes and the downregulation of antiapoptotic genes. 

### 3.13. GO-PtNPs Increase the Levels of 8-oxodG and 8-oxo-G by Causing Oxidative Damage to DNA 

Overproduction of ROS can induce lipid peroxidation as well as 3-( pyrimido[1,2-α]purin-10(3H)-one and 8-oxodG. Nanoparticles cause several toxic effects, which include chromosomal aberrations, DNA strand breakage, oxidative damage to DNA, and mutations [82,83]. Several assays have been used for the detection of NP-related oxidative damaged to DNA, especially measuring the oxidative level of 8-oxodG. Among several oxidative stress markers, 8-oxodG plays a significant role in DNA damage due to the prevalent oxidative lesions. To measure the levels of 8-oxodG and 8-oxoG, LNCaP cells were treated with IC50 concentrations of GO, GO-PtNPs, and PtNPs for 24 h. Cells treated with GO and PtNPs displayed 4- and 7-fold increases in 8-oxodG and 8-oxoG levels, respectively, compared to those in the control group. Similarly, the cells treated with GO and PtNPs displayed 3-fold and 5-fold increases, respectively, compared to the control group. Cells treated with GO-PtNPs displayed a 12-fold increase compared to the control (Figure 13). 

In another study, the exposure of human cells to PtNPs increased DNA damage, accumulation of cells at the S-phase of the cell cycle, and apoptosis [77]. HEK cells exposed to PtNPs reportedly displayed dose-dependent DNA damage that could be severe [56]. In another study [84], Fe_3_O_4_-NPs induced significant generation of 8-oxodG only at a higher concentration (60 µg/mL) and after more than 24 h of incubation. LNCaP cells exposed to PtNPs are expected to display significantly increased intracellular levels of ROS, which will damage DNA. A recent study suggested that PtNPs inhibit DNA replication and affect the secondary structure of DNA at higher concentrations in human cells and bacteria. In the study, the encapsulation of PtNPs in liposomes (LipoPtNPs) caused approximately 2.4 times higher DNA damage in comparison with CisPt, LipoCisPt, and PtNPs [85]. This was also found in our system, where exposure of LNCaP cells was significantly associated with higher levels of both 8-oxodG and 8-oxoG adducts after 24 h of incubation in comparison to that in the untreated cells. The generation of oxidative DNA lesions seems to be responsible for the induction of apoptotic death in cancer cells.

### 3.14. Impact of GO-PtNPs on Expression of GENES involved in Cell Cycle and DNA Damage

To further substantiate that oxidative DNA damage induced apoptosis, we evaluated the effect of GO-PtNPs on cell cycle arrest and DNA damage by evaluating the expressions of repair genes, including CDK2, CDK4, GADD45A, OGG1, APEX1, CREB1, UNG, and POLB, by RT-PCR after 24 h exposure to GO, GO-PtNPs, and PtNPs. The genes were significantly upregulated from 1- to 4-fold. CDK2 and CDK4 are tightly regulated by p21, which is controlled by p53. Cdk2 and Cdk4 inhibition by p21 are significant in G_1_ arrest upon DNA damage by various stresses and for cellular senescence, and p21 is an essential gene for p53-mediated G1 arrest in human cancer cells [86,87]. p53 is also believed to inhibit Cdk4 activity through p21 and by the repression of Cdk4 synthesis [88,89]. Inhibition of the cyclin-dependent kinases Cdk2 and Cdk4 is initiated by p53 and p21. It was reported [78] that the effect of PtNPs was more potent than that of cisplatin in hepatocellular carcinoma induced in rats. PtNPs inhibit cell proliferation by the induction of apoptotic cell death, which reduces cell viability and causes internucleosomal DNA fragmentation, G2/M cell cycle arrest, and hypodiploid accumulation [90]. The release of platinum ions from PtNPs inhibits cell division by binding to DNA, which causes DNA damage and downregulates the expression of proliferating cell nuclear antigen [77]. Furthermore, the present results suggested that GO, GO-PtNPs, and PtNPs significantly induced the expression of all the tested genes. In particular, the upregulation by GO-PtNPs was 2- to 4-fold higher than that in cells treated with GO or PtNPs (Figure 14). 

Graphene quantum dots trigger ROS generation, which triggers upregulation of genes associated with DNA damage [91]. In another study, GO-treated cells showed increased expression of DNA damage genes, including ATM and RAD51 [92]. Thus, GO-PtNPs effectively induce apoptosis compared to GO or PtNPs through inducing oxidative stress in LNCaP cells. These results agree with those of previous studies on the effect of PtNPs on fibroblast, glioblastoma, and A549 lung carcinoma cells [53,77]. Additionally, the increased expressions of all tested genes, 8-oxo-dG, and 8-oxoG is suggestive of oxidative damage to DNA. A hypothetical model demonstrates that the mechanism of GO-PtNPs induced oxidative stress and DNA damage in LnCaP cells (Figure 15).

## 4. Conclusions

In this study, we have successfully demonstrated a simple, environmentally friendly, and green approach for synthesis of GO-PtNPs using vanillin as a reducing and stabilizing agent. Cytotoxicity of GO-PtNPs was explored by determinations of cell viability, cell proliferation, and series of cellular assays. The results revealed a more effective dose-dependent effect caused by GO-PtNPs compared to either GO or PtNPs. Assessment of cytotoxicity test by examination of lactate dehydrogenase leakage and membrane integrity revealed that GO-PtNPs potentially caused cell death. GO-PtNPs increased the level of reactive oxygen species, malondialdehyde (MDA), nitric oxide (NO), and carbonylated protein levels. The imbalance between pro- and antioxidant levels led to the loss of mitochondrial integrity. In addition, GO-PtNP-treated cells exhibited mitochondrial-mediated apoptosis due to the upregulation of p53, p21, Bax, and Bak, and the downregulation of Bcl-2 and Bcl-xl. GO-PtNPs decreased the MMP and subsequently decreased the level of adenosine triphosphate production. Ultimately, GO-PtNPs caused programmed cell death by the upregulation of proapoptotic genes, including p53, p21, Bax, Bak, caspase 9, and caspase-3, and downregulation of the antiapoptotic marker genes Bcl-2 and Bcl-xl. These results further substantiated that GO-PtNPs nanocomposites can potentially disturb cell viability by inducing DNA cellular damage and genotoxicity by modulating the genes expression responsible for cell cycle arrest, DNA damage, and DNA repair, and increasing the levels of 8-oxo-dG and 8-oxo-G. GO-PtNPs significantly impaired the multiplication of cancer cells compared to GO or PtNPs alone. Hence, we can conclude that GO-PtNPs are potentially valuable as an alternate therapeutic agent for cancer. This piece of work could provide a step forward to improve therapeutic efficiency of biologically synthesized biocompatible and nontoxic agent seems to be one of promising techniques for cancer treatments.

## Figures and Tables

**Figure 1 polymers-11-00733-f001:**
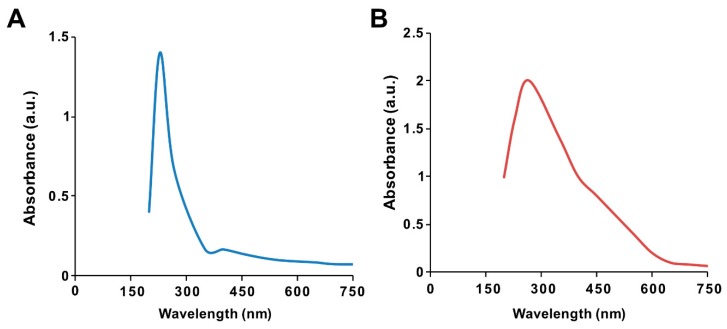
Synthesis and characterization of graphene oxide (GO) and graphene oxide–green platinum nanoparticles (GO-PtNPs). Ultraviolet–visible spectroscopy of GO (**A**) and GO-PtNPs (**B**). At least three independent experiments were performed for each sample and reproducible results were obtained.

**Figure 2 polymers-11-00733-f002:**
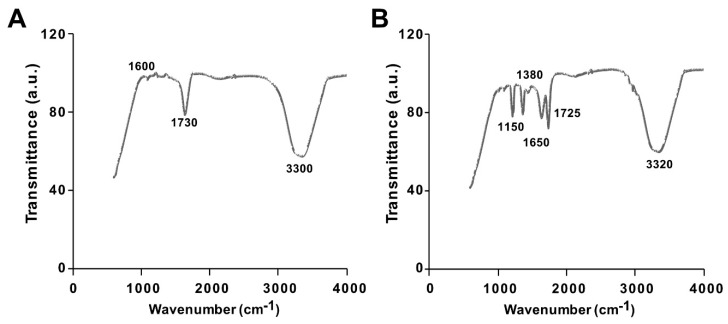
Characterization of GO and GO-PtNPs by Fourier-transform infrared spectroscopy (FTIR). FTIR images of GO (**A**) and GO-PtNPs (**B**). At least three independent experiments were performed for each sample and reproducible results were obtained.

**Figure 3 polymers-11-00733-f003:**
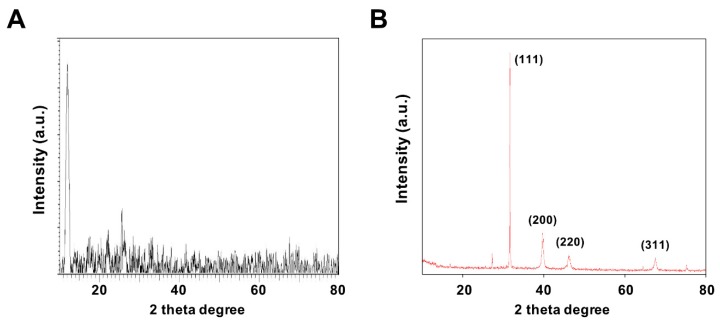
Characterization of GO and GO-PtNPs by XRD. XRD images of GO (**A**) and GO-PtNPs (**B**). At least three independent experiments were performed for each sample and reproducible results were obtained.

**Figure 4 polymers-11-00733-f004:**
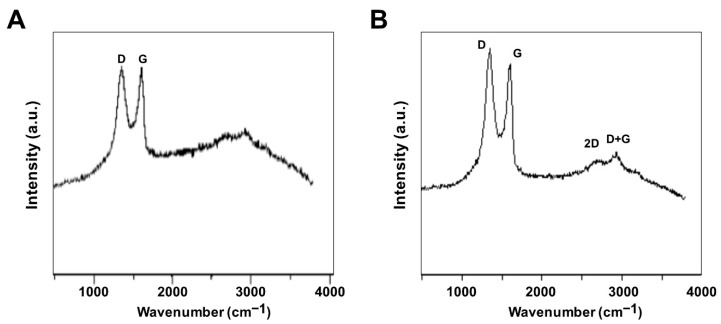
Characterization of GO and GO-PtNPs by Raman spectroscopy. Raman spectroscopy images of GO (**A**) and GO-PtNPs (**B**). At least three independent experiments were performed for each sample and reproducible results were obtained.

**Figure 5 polymers-11-00733-f005:**
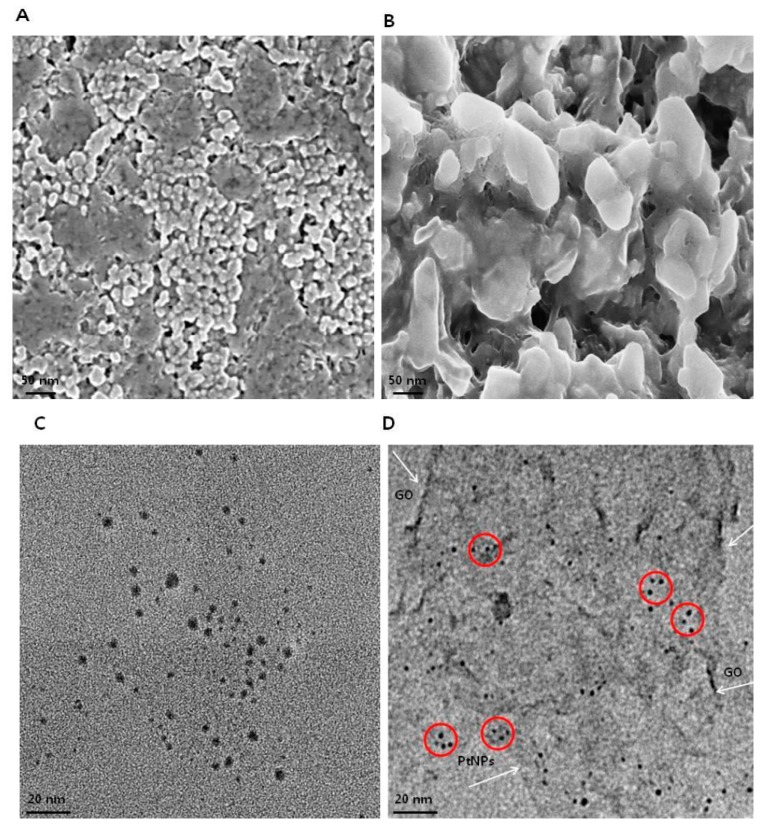
Characterization of GO and GO-PtNPs by SEM and TEM. Morphology of GO (**A**) and GO-PtNPs (**B**), and size of GO (**C**) and GO-PtNPs (**D**) were analyzed by SEM and TEM, respectively. The red circle indicates decoration of PtNPs particles on the surface of graphene sheet (White arrow). The graphene sheet depicted as wrinkled sheet-like structure.

**Figure 6 polymers-11-00733-f006:**
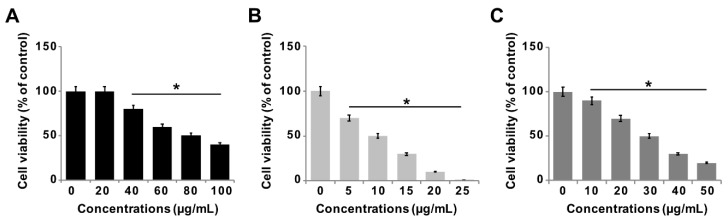
GO, GO-PtNPs, and PtNPs inhibit cell viability of LNCaP cells. The viability of LNCaP cells was determined after 24-h exposure to different concentrations of GO (**A**), GO-PtNPs (**B**), and PtNPs (**C**) using CCK-8. The treated groups showed statistically significant differences from the control group by the Student’s *t*-test (* *p* < 0.05).

**Figure 7 polymers-11-00733-f007:**
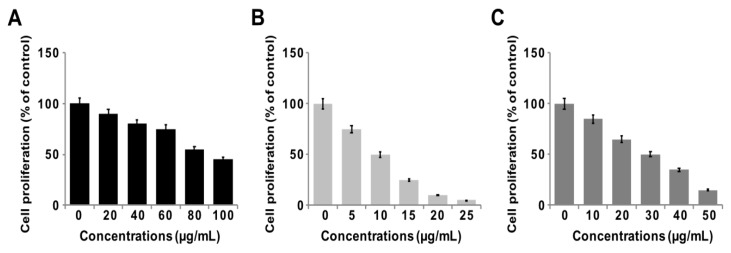
GO, GO-PtNPs, and PtNPs inhibit cell proliferation of LNCaP cells. Cell proliferation of LNCaP cells was determined using BrdU assay after 24-h exposure to different concentrations of GO (**A**), GO-PtNPs (**B**), and PtNPs (**C**). The treated groups showed statistically significant differences from the control group by the Student’s *t*-test (* *p* < 0.05).

**Figure 8 polymers-11-00733-f008:**
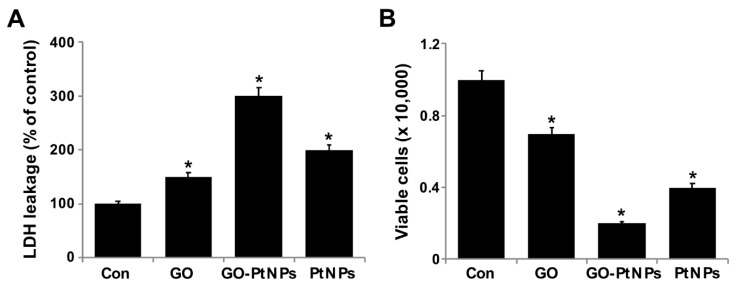
GO, GO-PtNPs, and PtNPs increase the leakage of LDH and cell death. LNCaP cells were treated with respective IC50 concentrations of GO, GO-PtNPs, and PtNPs for 24 h, and the LDH activity was measured at 490 nm using the LDH cytotoxicity kit (**A**). Cell death was determined by trypan blue assay after 24 h of exposure to GO, GO-PtNPs, and PtNPs for 24 h. Cell death was quantified by the ratio of living cells (**B**). The treated groups showed statistically significant differences from the control group by the Student’s *t*-test (* *p* < 0.05).

**Figure 9 polymers-11-00733-f009:**
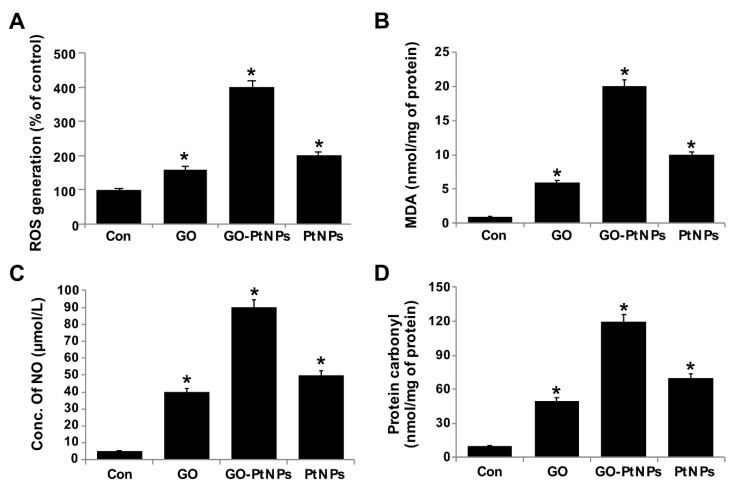
GO, GO-PtNPs, and PtNPs increase ROS generation (**A**), lipid peroxidation (**B**), and nitric oxide (**C**) and carbonylated protein content (**D**) in LNCaP cells. LNCaP cells was exposed to respective IC50 concentrations of GO, GO-PtNPs, and PtNPs for 24 h and then Spectrophotometric analysis of ROS was measured using DCFH-DA (**B**). The concentration of MDA was measured MDA using a thiobarbituric acid reactive substances assay and expressed as nanomoles per milliliter (**C**). NO production was quantified spectrophotometrically using the Griess reagent and expressed as micromoles per milliliter (**D**). Protein carbonyl content was measured and expressed relative to the total protein content. The treated groups showed statistically significant differences from the control group by the Student’s *t*-test (* *p* < 0.05).

**Figure 10 polymers-11-00733-f010:**
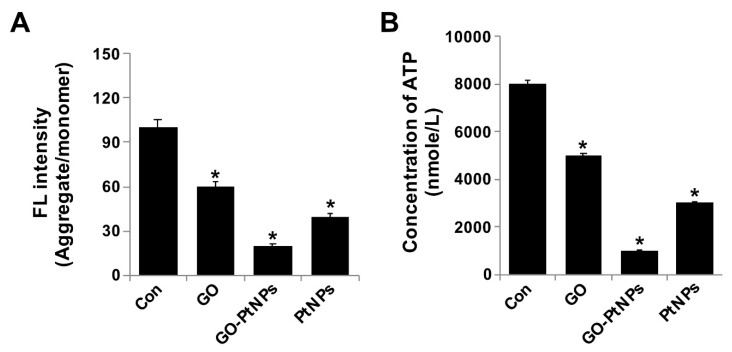
GO, GO-PtNPs, and PtNPs decrease mitochondrial membrane potential and ATP content. LNCaP cells were treated with respective IC50 concentration of GO, GO-PtNPs, and PtNPs for 24 h and spectrophotometric determination of JC-1 monomer/aggregate formation using cationic fluorescent indicator JC-1 (**A**). Intracellular ATP content (**B**). The treated groups showed statistically significant differences from the control group by the Student’s *t*-test (* *p* < 0.05).

**Figure 11 polymers-11-00733-f011:**
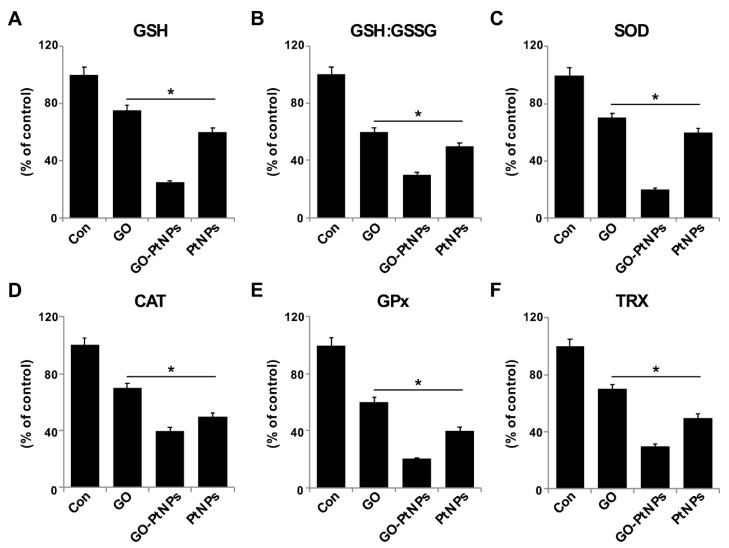
Effect of GO, GO-PtNPs, and PtNPs on antioxidant markers. LNCaP cells were treated with respective IC50 concentration of GO, GO-PtNPs, and PtNPs for 24 h. After incubation, cells were harvested and washed twice with an ice-cold phosphate-buffered saline solution. The cells were collected and disrupted by ultrasonication for 5 min on ice. GSH concentration is expressed as percentage of control (**A**). GSH:GSSG ratio is expressed as percentage of control (**B**). SOD concentration is expressed as percentage of control (**C**). CAT is expressed as percentage of control (**D**). GPx concentration is expressed as percentage of control (**E**). TRX is expressed as percentage of control (**F**). There was a significant difference in treated cells compared to untreated cells with Student’s *t*-test (* *p* < 0.05).

**Figure 12 polymers-11-00733-f012:**
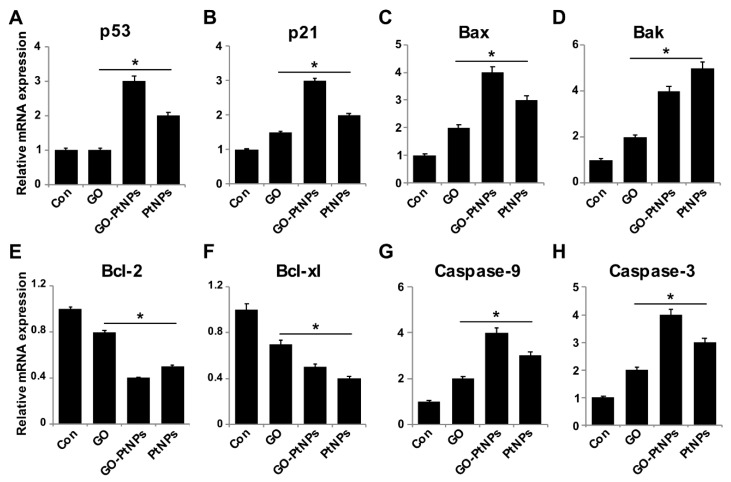
Effect of GO, GO-PtNPs, and PtNPs on the expression of pro- and antiapoptotic genes. LNCaP cells were treated with respective IC50 concentration of GO, GO-PtNPs, and PtNPs for 24 h. The relative messenger RNA(mRNA) expression of P53 (**A**), P21 (**B**), Bax (**C**), Bak (**D**) Bcl-2 (**E**), Bcl-xl (**F**), caspase-9 (**G**), and caspase-3 (**H**) was analyzed by quantitative reverse-transcription polymerase chain reaction in LNCaP cells treated for 24 h. After 24-h treatment, expression fold level was determined as fold changes in reference to expression values against GAPDH. Results are expressed as fold changes. There was a significant difference in treated cells compared to untreated cells with Student’s *t*-test (* *p* < 0.05).

**Figure 13 polymers-11-00733-f013:**
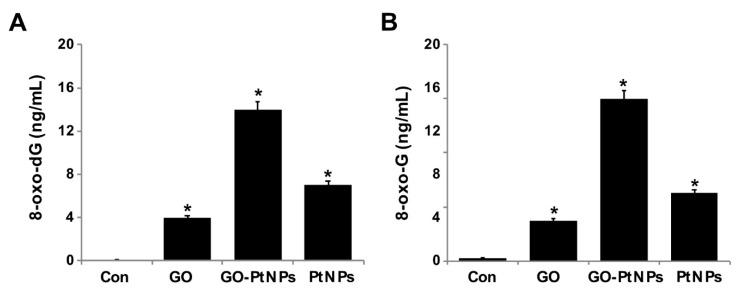
GO, GO-PtNPs, and PtNPs increase DNA damage. LNCaP cells were treated with respective IC50 concentration of GO, GO-PtNPs, and PtNPs for 24 h. 8-oxo-dG and 8-oxo-G were measured after 24 h of exposure of LNCaP cells. There was a significant difference in treated cells compared to untreated cells with Student’s *t*-test (* *p* < 0.05).

**Figure 14 polymers-11-00733-f014:**
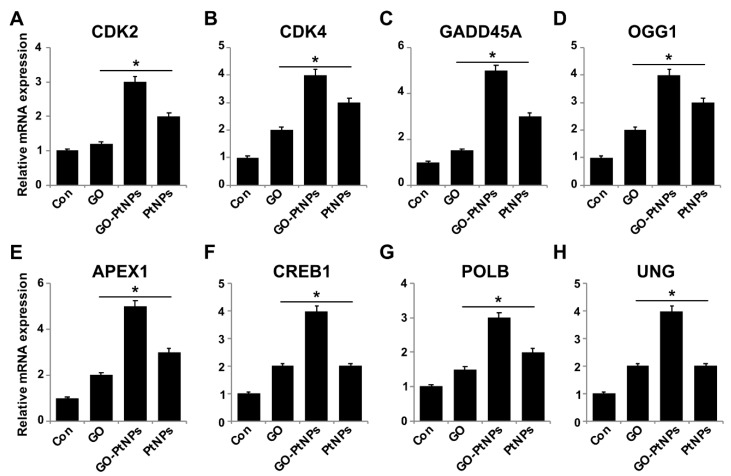
Effect of GO, GO-PtNPs, and PtNPs on expression cell cycle arrest and DNA damage genes. LNCaP cells were treated with respective IC50 concentrations of GO, GO-PtNPs, and PtNPs for 24 h. Relative messenger RNA(mRNA) expression of CDK2 (**A**), CDK4 (**B**), GADD45A (**C**), OGG1 (**D**) APEX1 (**E**), CREB1 (**F**), POLB (**G**), and UNG (**H**) was analyzed by quantitative reverse-transcription polymerase chain reaction in LNCaP cells treated for 24 h. After 24-h treatment, expression fold level was determined as fold changes in reference to expression values against GAPDH. Results are expressed as fold changes. There was a significant difference in treated cells compared to untreated cells with Student’s *t*-test (* *p* < 0.05).

**Figure 15 polymers-11-00733-f015:**
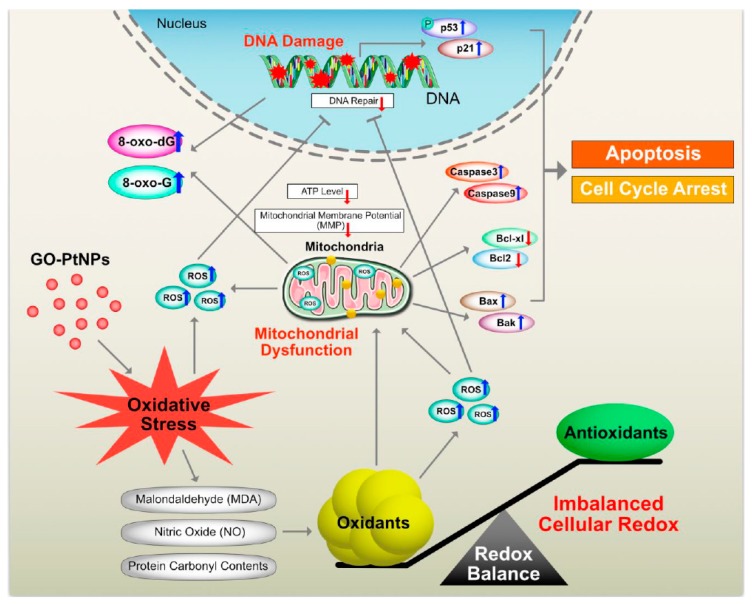
The hypothetical model demonstrates that the impact of GO-PtNPs on oxidative stress induced DNA damage in LnCaP cells.

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
