# Peer review of "Graphene Oxide–Platinum Nanoparticle Nanocomposites: A Suitable Biocompatible Therapeutic Agent for Prostate Cancer"

_polymers, 2019, doi:10.3390/polym11040733_

Round 1

Reviewer 1 Report

In general, the manuscript is solid in its content and significance, nonetheless some issues SHOULD be addressed before consideration for publication:      

- Claims including “It is estimated that there will be almost 1.3 million new cases of prostate cancer and 359,000 associated deaths worldwide in 2018, ranking as the second most frequent cancer and the fifth leading cause of cancer death in men (Bray et al., 2018). ”; “… however the treatment is depends on individual cases …”; “however it cause undesired side effects.”; and many others should be corrected.

-      There are many grammar errors and typos throughout the manuscript. They should be corrected. Besides, the English SHOULD be revised and corrected.

-       UPDATED and relevant references (One-Pot Synthesis of Reduced Graphene Oxide/Metal (Oxide) Composites. ACS Appl. Mater. Interfaces20179 (43), pp 37962–37971; Graphene Oxide/ZnS:Mn Nanocomposite Functionalized with Folic Acid as a Nontoxic and Effective Theranostic Platform for Breast Cancer Treatment. Nanomaterials 20188, 484; Graphene oxide–metal nanocomposites for cancer biomarker detection. RSC Adv., 2017,7, 35982-35991; Single-crystal γ-MnS nanowires conformally coated with carbon. ACS applied materials & interfaces 2014, 6, 1180-1186; Highly Sensitive Electrochemical Sensor for Anticancer Drug by a Zirconia Nanoparticle-Decorated Reduced Graphene Oxide Nanocomposite. ACS Omega. 2018 3, 14597–14605) should be included to support claims such as: “Graphene-based nanocomposites with metal nanoparticles show immense interest due to their extraordinary physical, chemical properties and biological properties.”; “The excellent properties of combination of graphene based materials and metal nanoparticles shows promising nanomaterial in many fields of application such as electronic-devices, sensors, nanocomposites, energy storage and supercapacitors”; and others SHOULD be included in the Introduction of the revised manuscript.

-      The exotic properties of graphene are translated to graphene oxide? This should be clarified in the Introduction.

-      The authors should state clearly and plainly in the last para of the Introduction with convincing arguments what is the role of graphene oxide and/or what will add to Pt as therapeutic agent?

-      In Figure 1, all panels containing GO and GO-PtNPs separately should be merged into individual panels, except TEM images.

-      HRTEM analysis should be added to Figure 1 to support the claims that Pt NPs are decorating the graphene sheet surface, and to claim that they are spherical and without agglomeration. Uniform decoration? Where is the evidence? Graphene sheets are considered nano? Where is the evidence? How thick they are (cross-section images are needed)? What about dimensions? These issues SHOULD be addressed with data.

-      Why the composite was not coated with any polymer to avoid toxicity and agglomeration? Drastic toxicity of bare GO is well documented, in that sense what is the point to study its toxicity?

-      Maybe the more drastic toxicity of the composite (heavier) is due to the accumulation onto the cells, any evidence on this?

-      It is not clear why the composite induces more damage to the cell membrane (or more ROS production) when compared to its counterparts. This should be mentioned and clearly explained at molecular level in the revised manuscript. 

-      From the PCR data, the “intrinsic pathway of apoptosis, which is mediated by the upregulation of pro-apoptotic genes and the downregulation of anti-apoptotic genes” is also observed at protein level (e.g. Western)? 

-      An illustration of the potential “oxidative damage to DNA” pathway SHOULD be included in the revised manuscript in the form of schematics.

Author Response

Response to the reviewer comments-1

We immensely thank the reviewer for their valuable and constructive comments that greatly facilitated us for improving the overall quality of the manuscript. As per the reviewer constructive comments, the corrections were carried out in the manuscript. We hopefully believe that we have addressed all the comments mentioned by the reviewers carefully and precisely. All the changes are highlighted in yellow color in the revised manuscript. In addition, this manuscript was proof read by native English speaker by Editage editing company, Seoul, South Korea.

Comments and Suggestions for Authors

In general, the manuscript is solid in its content and significance; nonetheless some issues SHOULD be addressed before consideration for publication:

Thanks to the reviewer for constructive and positive response about our manuscript and also further comments to improve overall quality of the manuscript.

Claims including “It is estimated that there will be almost 1.3 million new cases of prostate cancer and 359,000 associated deaths worldwide in 2018, ranking as the second most frequent cancer and the fifth leading cause of cancer death in men (Bray et al., 2018). however the treatment is depends on individual cases …”; “however it cause undesired side effects.”; and many others should be corrected.

Thanks to the reviewer for indicating incorrect sentences and also repetitive sentences. We deleted ambiguous sentences in the revised manuscript.  

There are many grammar errors and typos throughout the manuscript. They should be corrected. Besides, the English SHOULD be revised and corrected.

Thanks to the reviewer for critical evaluation. We absolutely agree with reviewer for many errors. The revised manuscript was proof read by native English speaker by Editage editing company, Seoul, South Korea.

UPDATED and relevant references (One-Pot Synthesis of Reduced Graphene Oxide/Metal (Oxide) Composites. ACS Appl. Mater. Interfaces, 2017, 9 (43), pp 37962–37971; Graphene Oxide/ZnS:Mn Nanocomposite Functionalized with Folic Acid as a Nontoxic and Effective Theranostic Platform for Breast Cancer Treatment. Nanomaterials 2018, 8, 484; Graphene oxide–metal nanocomposites for cancer biomarker detection. RSC Adv., 2017,7, 35982-35991; Single-crystal γ-MnS nanowires conformally coated with carbon. ACS applied materials & interfaces 2014, 6, 1180-1186; Highly Sensitive Electrochemical Sensor for Anticancer Drug by a Zirconia Nanoparticle-Decorated Reduced Graphene Oxide Nanocomposite. ACS Omega. 2018 3, 14597–14605) should be included to support claims such as: “Graphene-based nanocomposites with metal nanoparticles show immense interest due to their extraordinary physical, chemical properties and biological properties.”; “The excellent properties of combination of graphene based materials and metal nanoparticles shows promising nanomaterial in many fields of application such as electronic-devices, sensors, nanocomposites, energy storage and supercapacitors”; and others SHOULD be included in the Introduction of the revised manuscript.

Thanks to the reviewer for great input. According to the reviewer suggestions we included all the references in the revised manuscript.

The exotic properties of graphene are translated to graphene oxide? This should be clarified in the Introduction. The authors should state clearly and plainly in the last para of the Introduction with convincing arguments what is the role of graphene oxide and/or what will add to Pt as therapeutic agent?

Thanks to the reviewer for excellent suggestions. According to the reviewer comments we included the role of graphene oxide and platinum nanoparticles and combination of both in the section of introduction.

In Figure 1, all panels containing GO and GO-PtNPs separately should be merged into individual panels, except TEM images.

We absolutely agree with reviewer comments. For readership convenience and clear understanding, we represented GO and GO-PtNPs as individual panel. Now we improved the quality of figures and alignment of figures in the revised manuscript.

HRTEM analysis should be added to Figure 1 to support the claims that Pt NPs are decorating the graphene sheet surface, and to claim that they are spherical and without agglomeration. Uniform decoration? Where is the evidence? Graphene sheets are considered nano? Where is the evidence? How thick they are (cross-section images are needed)? What about dimensions? These issues SHOULD be addressed with data.

Thanks to the reviewer for clarification, response to your first question we have already included HRTEM analysis of PtNPs and PtNPs decorated on the surface of graphene sheets as figure 1K and L. To show the clarity of figures, we analyzed new samples and included new figures.

Response to your second question, graphene oxide nanoparticles considered to be nanoparticles (size between 1-100 nm) depend son size of the sheets. In our experiment setup, the prepared graphene nanosheets showed with an average size between 20 and 50 nm. Therefore it can be considered as nanoparticles. The size of the dimension was included in the figure.

Why the composite was not coated with any polymer to avoid toxicity and agglomeration? Drastic toxicity of bare GO is well documented, in that sense what is the point to study its toxicity?

Thanks to the reviewer for thought provoking question. We absolutely agree with reviewer bare GO shows significant toxicity against both cancer and non-cancer cells. Our aim of study is to determine the effect of simultaneous reduction of graphene oxide and platinum ions in a single platform against cancer cells using biological molecule like vanillin as reducing and stabilizing agent. In our experimental setup we used vanillin as reducing agent and stabilizing agent to avoid toxicity and aggregation. The synthesized composite materials show significant biocompatibility and excellent anticancer properties.

Maybe the more drastic toxicity of the composite (heavier) is due to the accumulation onto the cells, any evidence on this?

Thanks to the reviewer excellent idea. The toxicity of composite is not related weight which is merely depends on size, charge, chemical composition, oxidation state and so on. The two different combination of materials exhibited different physical and chemical properties and excellent toxicity against cancer cells.

It is not clear why the composite induces more damage to the cell membrane (or more ROS production) when compared to its counterparts. This should be mentioned and clearly explained at molecular level in the revised manuscript.

In general nanoparticles such as carbon silver, platinum, palladium ions bind to protein disulfide bonds in the cytoplasm, causing deformities in the protein structure. These malformed proteins are then incorporated into the plasma membrane, leading to alterations in cell permeability and cellular death. In addition, graphene induce toxicity owing to its distinct physicochemical characteristics such as purity, lateral dimension, size of the sheets, and oxidation-state, which may influence its cellular uptake, biodegradation and toxicity. Once in contact with the cell membrane, graphene sheets can create an impermeable encasement affecting the normal exchange between the cell and the extracellular environment, Graphene oxide can also damage the cell membrane through strong electrostatic interactions between the negatively charged oxygen groups on its surface and the positively charged lipids present on cell membranes. All these molecular level cause of cell death were included in the revised manuscript.

From the PCR data, the “intrinsic pathway of apoptosis, which is mediated by the upregulation of pro-apoptotic genes and the downregulation of anti-apoptotic genes” is also observed at protein level (e.g. Western)? 

Thanks to the reviewer for enquiring confirmative test. Editor and other reviewers given short period of time (within 10 days) to resubmit the revised manuscript, therefore ordering chemicals and getting data would take long time and also this manuscript already contains extensive data. Furthermore, previously several studies have been confirmed that including from our group, graphene oxide nanocomposites increase the expression of proapoptotic protein levels and down regulate anti-apoptotic protein levels in various type of cancer cells.

An illustration of the potential “oxidative damage to DNA” pathway SHOULD be included in the revised manuscript in the form of schematics.

According to the reviewer comments we included the potential oxidative damage to DNA pathway in the revised manuscript as figure 11.

Once again thanks to the reviewer for wonderful comments to improve the overall quality of the manuscript.

Reviewer 2 Report

The authors synthesized PtNPs using vanillin and then prepared and characterized GO-PtNPs. They stated that GO-PtNPs are both cytotoxic and genotoxic. The authors demonstrated that GO-PtNPs have potential as an alternate and effective cancer therapeutic agent. This is interesting research. However, there are some minor revisions required before the publication.

1. The SEM and TEM images for the nanoparticles are not clear, try to use some better image or improve the resolution.

2. The whole paper is too redundant, especially the abstract and the conclusion. It is not easy for audiences to follow the ideas clearly.

Author Response

Response to the reviewer comments-2

Comments and Suggestions for Authors

We immensely thank the reviewer for their valuable and constructive comments that greatly facilitated us for improving the overall quality of the manuscript. As per the reviewer constructive comments, the corrections were carried out in the manuscript. We hopefully believe that we have addressed all the comments mentioned by the reviewers carefully and precisely. All the changes are highlighted in yellow color in the revised manuscript. In addition, this manuscript was proof read by native English speaker by Editage editing company, Seoul, South Korea.

The authors synthesized PtNPs using vanillin and then prepared and characterized GO-PtNPs. They stated that GO-PtNPs are both cytotoxic and genotoxic. The authors demonstrated that GO-PtNPs have potential as an alternate and effective cancer therapeutic agent. This is interesting research. However, there are some minor revisions required before the publication.

Thanks to the reviewer for encouraging, positive and constructive comments to improve overall quality of the manuscript.

1. The SEM and TEM images for the nanoparticles are not clear, try to use some better image or improve the resolution.

According to the reviewer comments we performed new experiments and included new figures and also improved quality of all figures in the revised manuscript.

2. The whole paper is too redundant, especially the abstract and the conclusion. It is not easy for audiences to follow the ideas clearly.

Thanks to the reviewer for critical observation. According to the reviewer comments, first of all we deleted redundant sentences and also we rectified the error observed in the abstract and conclusion.

Once again thanks to the reviewer for wonderful comments to improve the overall quality of the manuscript.

Reviewer 3 Report

Authors present synthesis and studies on graphene oxide – platinum nanoparticles nanocomposites. Mentioned nanocomposites have been analyzed mainly in viewpoint of their future application in anticancer therapy. Due to the fact that nanotechnology is currently one of the leading areas that seems to be perspective for application for biomedical purposes, the research subject of the paper is undoubtedly worth investigating. However, manuscript should be significantly improved to be acceptable for publication and needs major revisions.

Section “Materials” should be re-written and ordered. Authors need to sort the information concerning the origin of all applied reagents.

Hummers’ method as well as its modification mentioned in Section 2.2.: should be briefly described by Authors.

Authors mentioned that the reaction of preparation of GO-PtNPs was stopped after 12 h and that after washing with water the pure product was obtained. On what basis was it concluded that the reaction should be stopped? Were there any changes in the color of the reaction mixture etc.?

Authors mentioned in line 139-140 that: “All the samples were characterized as prescribed previously [2, 4, 5]”. How the samples were characterized?

Authors should briefly describe the procedure of the cell proliferation assay conducted using BrdU. What about its incorporation by tested cells?

Authors used many abbreviations that sometimes have not been explained (e.g. IC50) or have been explained in not suitable section; e.g. abbreviations such as NO, ROS and MD  have been firstly used in section 2.7. and explained for the first time in section 3.5. It should be corrected.

Authors many times used the following sentence “…it was described previously..” (e.g. 173, 185 line etc.) and then the proper reference is added. In such a situation, the mentioned information should be briefly described in the manuscript.

Quality of all figures must be significantly corrected. Figure 1. needs to be divided into few figures presenting results of particular analyses separately.

Additionally, in the case of FT-IR spectra- there are no units on axes; the range of x-axis should be as follows: 4000 – 700 (or 500) cm-1; all peaks presented in Figure 1D should be described.

Figure 1A-1B- the range of x-axis should be changed, e.g. 150 – 750 nm.

Captions of Figures are too long and extensive and should be improved.

Conclusions of the manuscript should be re-written. They need to present major results of conducted studies without introduction of too many specific abbreviations.

Section “References” needs to be corrected and prepared according to the recommendations of the Journal (e.g. abbreviations of the titles instead of the whole ones). Furthermore, references in the Introduction need to be given in square brackets.

Manuscript should also be significantly improved grammatically and linguistically.

Author Response

Response to the reviewer comments-3

We immensely thank the reviewer for their valuable and constructive comments that greatly facilitated us for improving the overall quality of the manuscript. As per the reviewer constructive comments, the corrections were carried out in the manuscript. We hopefully believe that we have addressed all the comments mentioned by the reviewers carefully and precisely. All the changes are highlighted in yellow color in the revised manuscript. In addition, this manuscript was proof read by native English speaker by Editage editing company, Seoul, South Korea.

Comments and Suggestions for Authors

Authors present synthesis and studies on graphene oxide – platinum nanoparticles nanocomposites. Mentioned nanocomposites have been analyzed mainly in viewpoint of their future application in anticancer therapy. Due to the fact that nanotechnology is currently one of the leading areas that seems to be perspective for application for biomedical purposes, the research subject of the paper is undoubtedly worth investigating. However, manuscript should be significantly improved to be acceptable for publication and needs major revisions.

Thanks to the reviewer for encouraging and positive responses and also further comments to improve overall quality of the manuscript.

Section “Materials” should be re-written and ordered. Authors need to sort the information concerning the origin of all applied reagents.

Thanks to the reviewer for excellent observation. We rewritten and reordered wherever it is necessary in the materials and method section and also we sort out the information concerning the origin of all applied reagents and also included detailed methdology.

Hummers’ method as well as its modification mentioned in Section 2.2.: should be briefly described by Authors.

According to reviewer comments, we briefly discussed about Hummers method in the revised manuscript.

Authors mentioned that the reaction of preparation of GO-PtNPs was stopped after 12 h and that after washing with water the pure product was obtained. On what basis was it concluded that the reaction should be stopped? Were there any changes in the color of the reaction mixture etc.?

Thanks to the reviewer for critical reading of the manuscript. Yes, we stopped the reaction after 12h due to color change from black to brown.

Authors mentioned in line 139-140 that: “All the samples were characterized as prescribed previously [2, 4, 5]”. How the samples were characterized?

Thanks to the reviewer for clarification. The characterizations of samples were included in the revised manuscript.

Authors should briefly describe the procedure of the cell proliferation assay conducted using BrdU. What about its incorporation by tested cells?

Thanks to the reviewer for thought-provoking comments. According to the reviewer comments we included incorporation of BrdU into the newly synthesized DNA in the revised manuscript.

Authors used many abbreviations that sometimes have not been explained (e.g. IC50) or have been explained in not suitable section; e.g. abbreviations such as NO, ROS and MD  have been firstly used in section 2.7. and explained for the first time in section 3.5. It should be corrected.

Response to your first question, thanks to the reviewer for critical evaluation of the manuscript. We included the definition for IC 50 in the cell viability section.

Response to your second question, we included abbreviations for ROS, MDA and NO included in the section 2.7 appeared in the first place rather than 3.5

Authors many times used the following sentence “…it was described previously..” (e.g. 173, 185 line etc.) and then the proper reference is added. In such a situation, the mentioned information should be briefly described in the manuscript.

Thanks to the reviewer for logical comments. According to the reviewer comments we included detailed methodology and also cited proper references in materials and method section.

Quality of all figures must be significantly corrected. Figure 1. needs to be divided into few figures presenting results of particular analyses separately. Additionally, in the case of FT-IR spectra- there are no units on axes; the range of x-axis should be as follows: 4000 – 700 (or 500) cm-1; all peaks presented in Figure 1D should be described.

Thanks to the reviewer for technical comments. Response to your first question, we agree with reviewer comments. For the quick understanding of readership all the characterization data represented as figure 1. According to the reviewer’s comments, we have rearranged the figures and improved the quality of all figures.

Response to your second question all the peaks appeared in figure 1D explained.

Figure 1A-1B- the range of x-axis should be changed, e.g. 150 – 750 nm.

Thanks to the reviewer to make clarity for readership, we changed the x-axis value from 200 to 150 nm.

Captions of Figures are too long and extensive and should be improved.

Thanks to the reviewer for observation of each and every sentence in each figure. In order to easy understanding for the readership, we included necessary information and we deleted repetitive information in the caption.

Conclusions of the manuscript should be re-written. They need to present major results of conducted studies without introduction of too many specific abbreviations.

Thanks to the reviewer for given importance to the conclusion. We modified the conclusion section with important messages and given definition to abbreviations.

Section “References” needs to be corrected and prepared according to the recommendations of the Journal (e.g. abbreviations of the titles instead of the whole ones). Furthermore, references in the Introduction need to be given in square brackets.

We apologize for the error. According to the reviewer comments we rectified the error in all references.

Manuscript should also be significantly improved grammatically and linguistically.

The revised manuscript was proof read by native English speaker by Editage editing company, Seoul, South Korea.

Once again thanks to the reviewer for wonderful comments to improve the overall quality of the manuscript.

Round 2

Reviewer 1 Report

The authors have successfully addressed the issues.

Author Response

Response to the reviewer -1

Comments and Suggestions for Authors

The authors have successfully addressed the issues.

Thanks to the reviewer for positive, encouraging and motivated comments

Reviewer 3 Report

Authors significantly improved proposed paper. Section “Materials” has been improved. Authors supplemented manuscript with all missing information. Brief descriptions of applied procedures have been added, all abbreviations have been explained. Quality of figures has been improved. Section “Conclusion” has also been changed. Majority of the suggestions of the Reviewer has been considered. However, paper still requires some minor revisions. Authors should re-write section “References” and prepare it according to the requirements of the Journal- e.g. use abbreviated journal titles instead of the whole ones. Additionally, it will be better to divide Figure 1. into few figures (e.g. Fig. 1- UV-Vis analysis results; Fig. 2. – FT-IR spectra etc.) as it was previously suggested. After these small changes manuscript may be acceptable for publication in the Journal.

Author Response

Response to the reviewer comments -3-RII

Comments and Suggestions for Authors

Authors significantly improved proposed paper. Section “Materials” has been improved. Authors supplemented manuscript with all missing information. Brief descriptions of applied procedures have been added, all abbreviations have been explained. Quality of figures has been improved. Section “Conclusion” has also been changed. Majority of the suggestions of the Reviewer has been considered. However, paper still requires some minor revisions. Authors should re-write section “References” and prepare it according to the requirements of the Journal- e.g. use abbreviated journal titles instead of the whole ones. Additionally, it will be better to divide Figure 1. into few figures (e.g. Fig. 1- UV-Vis analysis results; Fig. 2. – FT-IR spectra etc.) as it was previously suggested. After these small changes manuscript may be acceptable for publication in the Journal.

We immensely thank the reviewer for their valuable and constructive comments that greatly facilitated us for improving the overall quality of the manuscript. As per the reviewer constructive comments, the corrections were carried out in the manuscript. According to the reviewer comments, we rectified the error in the reference section.

Response to your second comment, we divided figure 1 into five figures and increased resolution also to understand the figure easily.

Once again thanks to the reviewer for wonderful comments to improve the overall quality of the manuscript.
